# The Impact of the COVID-19 Pandemic on the Assessment of Healthcare and Anxiety Disorders in Patients with Chronic Diseases

**DOI:** 10.3390/ijerph20206956

**Published:** 2023-10-21

**Authors:** Edyta Cichocka, Anna Maj-Podsiadło, Janusz Gumprecht

**Affiliations:** Department of Internal Medicine, Diabetology and Nephrology in Zabrze, Medical University of Silesia, 40-055 Katowice, Poland; annamaj@sum.edu.pl (A.M.-P.); jgumprecht@sum.edu.pl (J.G.)

**Keywords:** anxiety, COVID-19, hemodialysis, diabetes, lockdown

## Abstract

(1) Background: We compared the impact of the COVID-19 pandemic on the functioning and mental health of chronically ill patients, namely those with hemodialysis (HD) and diabetes (DM). (2) Methods: We used a questionnaire to collect the medical data and the Generalized Anxiety Questionnaire (GAD-7) to measure the mood status. (3) Results: In both groups, a similar percentage of patients had a past COVID-19 infection and similar opinions about pandemic-related inconveniences. The most significant limitations of the study included mask wearing and the restriction of social contact. Mental disorders were significantly more frequently reported in the DM group. Sleep problems were found in approximately 30% of patients. Approximately 20% of patients in both groups declared that the pandemic had negatively affected the quality of their sleep. The mean score of the GAD-7 scale in the HD group did not differ according to gender. In the group of DM patients, a significant difference was observed between men and women, with women scoring higher compared to men. In both groups, the percentage of patients with GAD-7 scores > 5, > 10 and > 15 did not differ significantly. (4) Conclusions: In both groups, chronically ill patients reported anxiety disorders with similar frequency. In the DM group, more severe anxiety disorders were found in women. Mental disorders were significantly more prevalent in DM patients. It seems that HD patients coped better with the psychological aspects of pandemic-related stress and limitations.

## 1. Introduction

The global COVID-19 pandemic significantly changed the social life of people worldwide. Restrictions, lockdowns, emergency states and isolation may have led to mental changes and the onset of anxiety and depressive symptoms. Pandemic-related restrictions also significantly affected the functioning of and access to healthcare. A significant reduction in the number of outpatient visits, the transformation of some hospitals into departments responsible for the treatment of COVID-19 patients, and a reduction and delay in diagnostic admissions and elective procedures, including oncological surgery, resulted in a deterioration in medical care, which increased overall mortality. The adverse impact of the pandemic was particularly evident in patients with chronic diseases such as obesity, diabetes, chronic kidney disease and CVD, and oncology patients, whose prognosis and chances of survival were closely related to the continuation of treatment and often the need for hospitalization or surgery.

Uncertainty about the possibility of further treatment, the increased risk of developing and/or a severe course of COVID-19 are significant factors worsening the well-being of patients with many chronic conditions, thus contributing to increased anxiety and affecting their quality of life.

Studies on the effects of the COVID-19 pandemic on mental health have shown the detrimental psychological consequences of the SARS-CoV-2 pandemic outbreak in various patient populations [1,2]. However, only some studies have described the impact of the pandemic on the functioning and mental health of chronically ill patients.

Among the patients with chronic diseases, patients with end-stage kidney disease (ESKD) treated with renal replacement therapy based on repeated hemodialysis (HD) are a specific group. On the one hand, chronic kidney disease (CKD) significantly increases the risk of cardiovascular events and the risk of death, which is further exacerbated by COVID-19 [3]. On the other hand, during the pandemic, dialysis patients could not adhere to isolation or contact restriction, including in health care facilities, to minimize the risk of SARS-CoV-2 infection. Patients had to undergo regular hemodialysis or other medical procedures related to their disease [4].

Patients with type 2 diabetes mellitus (DM) are another group of patients with a chronic disease. In this group, the cardiovascular mortality rate, especially in patients with atherosclerotic cardiovascular disease (ASCVD), heart failure, or CKD, is several times higher compared to the population without DM. Poorly controlled type 2 DM leads to chronic complications, including CKD and the need for renal replacement therapy in the end stage of the disease.

The COVID-19 pandemic and its associated restrictions on access to outpatient clinics, including diabetes healthcare, translated into a reduction in the diagnosis of new cases of DM and the quality of care for diabetes patients. Online visits, which often did not provide insights into the current glycemic control needed and were limited to prescribing medications only, also contributed to the deterioration in glycemic control in patients with type 2 DM. In addition, consistent data published at the beginning of the pandemic showed that long-term, poorly controlled type 2 DM significantly increased the risk of death from SARS-CoV-2 infection [5], thus being an additional risk factor for this patient population.

The aim of this study was to investigate and compare the impact of the COVID-19 pandemic on the functioning and mental health of two groups with chronic diseases, i.e., chronic dialysis patients and type 2 DM subjects. We selected patients that differed from other chronically ill patients primarily in their treatment modality and prognosis for survival due to their dramatically high cardiovascular risk. We wanted to assess whether differences due to the specificity of these groups of patients could translate into the presence of anxiety disorders and other aspects of functioning during the COVID-19 pandemic.

## 2. Materials and Methods

The study was multicenter and included 244 hemodialysis (HD) patients from the Silesian region, Poland, who were recruited in the first three months of 2022. The data of a total of 244 patients were obtained. Another group consisted of patients with type 2 DM who underwent hospitalization in the first three months of 2022 due to poor glycemic control at the Clinical Hospital No. 1 in Zabrze, Poland. The data were obtained from 175 patients. The study was questionnaire-based and the Bioethics Committee of the Medical University of Silesia was informed about the study. 

In the group of DM patients, we excluded patients with newly diagnosed diabetes. We analyzed only patients who required the intensification of diabetes treatment (for this reason, they were hospitalized in the Diabetes Department) and most of them were treated with insulin or insulin with oral medications.

The patients were asked to complete a questionnaire that was prepared for the study. All patients gave their written informed consent. The inclusion criteria were as follows: age > 18 years, consent to participate in the study, the presence of DM for at least a year (DM group), or hemodialysis for at least 6 months (HD group). The exclusion criteria were as follows: a lack of consent to participate in the study or an inability to complete the questionnaire independently due to intellectual disorders and active oncological conditions.

The survey for HD patients consisted of four parts. The first part included sociodemographic data, and the second part asked questions about the duration of dialysis, comorbidities, hospitalizations in the past two years, access to specialists other than a nephrologist and problems related to obtaining prescriptions for chronic medications, and data on individuals’ current body weight and their weight before the COVID-19 pandemic. Additionally, the percentage of patients eligible for kidney transplantation was assessed. The third part included questions related to the COVID-19 pandemic (i.e., issues associated with the development and course of the SARS-CoV-2 infection, vaccination against COVID-19, and inconveniences due to the restrictions implemented in connection with the pandemic). The final part included questions on patients’ mental condition, particularly anxiety and sleep problems. Patients were asked to complete the Generalized Anxiety Questionnaire (GAD-7).

The survey for DM patients was prepared in a similar manner. The first part contained sociodemographic data. The second part included questions about the duration and treatment of DM, the chronic complications of DM, comorbidities, access to a diabetologist and other medical specialists, and problems related to obtaining prescriptions for chronic medications. Patients were asked about their current body weight and their levels of HbA1c before and after the COVID-19 pandemic (the current results were obtained during hospitalization). They were also asked about the difference between outpatient and telephone visits, and their preferences regarding the type of visit. The third part included questions related to the COVID-19 pandemic (questions about the development and course of the SARS-CoV-2 infection, vaccination against COVID-19, and inconveniences due to restrictions implemented in connection with the pandemic). The final part consisted of questions related to mental functioning, particularly anxiety and sleep problems. The patients were asked to complete the GAD-7. An additional question on anxiety relating to COVID-19 infection was added to the questionnaire in both study groups.

The questionnaires that were conducted among the patients differed between the HD and DM groups only in terms of questions about the specific treatment of the underlying disease—questions related to hemodialysis or the difficulties of transportation to dialysis.

### 2.1. Generalized Anxiety Questionnaire (GAD-7)

The questionnaire we used was a screening tool developed to determine the level of anxiety and assess the risk of generalized anxiety disorder (GAD). It consists of 7 items and is based on a four-point Likert scale. Item scores range from 0 to 3, depending on the frequency of the occurrence of a given phenomenon over the past 14 days (0—not at all, 1—several days, 2—more than half the days, 3—nearly every day). The questions in the questionnaire allowed us to assess anxiety, tension, nervousness, the ability to control these feelings, the ease with which the above symptoms developed, and problems with relaxation. A score of 5, 10, and 15 points showed the presence of mild, moderate and severe anxiety, respectively. A score of at least 10 points indicated a high probability of GAD [6]. An additional question was also assigned a score from 0 to 3 points.

### 2.2. Statistical Analysis

Data were presented as the number of cases with a percentage for qualitative variables. Quantitative variables were presented as the median of the first and third quartiles. To assess the normality of the variables, we used a histogram and a quantile–quantile (QQ) plot. Non-parametric tests were used to assess the significance of differences between the variables due to deviations from the normal distribution (Mann–Whitney U test to compare two independent samples, paired Wilcoxon test for dependent samples, and the Kruskal–Wallis test to compare many groups). Correlations were assessed using the Spearman’s rank correlation coefficient. The comparison of qualitative variables was based on Pearson’s chi-squared test. The analysis was performed using the R language in the RStudio environment [7] using tidyverse and janitor packages [8].

## 3. Results

### 3.1. Study Population

A total of 399 patients were eligible for analysis. The sociodemographic data of the participants are given in Table 1. The study patients were of similar age (HD group: mean 63.7 ± 15.4 years; DM group: mean 65.4 ± 10.3 years). Men were predominant among the HD patients and accounted for 55.8% of patients, while in DM patients, men accounted for 40%. DM patients were married significantly more often (69% vs. 57.1%). Nearly 80% of patients in both groups lived with their families. Significantly more DM patients were professionally active (27% vs. 12.5%). In the group of HD patients, 81% reported that their financial situation deteriorated during the pandemic compared to 85% of DM patients (*p* = 0.034). A similar percentage of patients expressed the opinion that the pandemic had changed their lives (68.3% vs. 67%).

### 3.2. Clinical Data

The mean duration of dialysis was 45 months (19.2–84). In total, 70% of patients had been on dialysis before the pandemic. The disease duration before dialysis was not considered in the group of HD patients. In total, 12% of patients were eligible for kidney transplantation. Difficulties related to dialysis (missed dialysis or arriving late for dialysis because of transport delays) affected a small percentage of patients (6.6% and 11.6%, respectively).

The mean duration of DM was 12.9 ± 8.31 years. In total, 47.4% of patients were treated with insulin and oral hypoglycemic drugs, 38.3% were treated with oral drugs only (at least two drugs require the intensification of therapy), and 14.3% were treated with insulin only.

The DM and HD patients presented with other medical conditions, such as hypertension (HD—85.6% vs. DM—82%). Patients with DM were diagnosed with hypercholesterolemia more, and HD patients were diagnosed with ischemic heart disease more (Table 2). In turn, HD patients presented with more comorbidities than DM patients.

One hundred and forty patients (65.63%) in the HD group had been hospitalized for various medical reasons in the past two years. Due to technical problems, the data on previous hospitalizations in DM patients were not collected.

In HD patients, the mean BMI was 26.8 kg/m^2^ during the follow-up and was significantly lower than the level before the pandemic (27.7 kg/m^2^). Patients with DM were obese (BMI > 30 kg/m^2^) and no differences were found between BMI values before the COVID-19 pandemic and at the time of the study. The level of HbA1c increased significantly over the two years of the pandemic (7.5% vs. 8.05%), which was consistent with the patients’ assessment with regard to poor glycemic control during the pandemic.

A similar percentage of patients in both groups preferred outpatient visits (HD—62% vs. DM—65%). In total, 10.2% of HD patients reported problems obtaining prescriptions during the COVID-19 pandemic compared to 3% of DM patients. Patients with DM were significantly less likely to have difficulty contacting other specialists compared to HD patients (34.2% vs. 46.6%).

### 3.3. COVID-19 Status

In both groups, a similar percentage of patients had experienced past COVID-19 infection (HD—75.4% vs. DM 71.4%) (Table 3). A severe course of disease (defined as the need for oxygen therapy, antibiotic therapy, and systemic steroid treatment) was reported by 24.5% of patients in the HD group and 18% of the DM group (*p* = 0.5436). Oxygen therapy was significantly more often applied to DM patients, while antibiotics were significantly more often administered to HD patients. In total, 85.7% of HD patients were vaccinated against SARS-CoV-2 compared to 77% of DM patients (*p* = 0.01286). A similar percentage of patients in both groups reported the death of a relative due to COVID-19.

Both groups had similar opinions about pandemic-related inconveniences, the most significant limitations being mask wearing (50.9% in HD group and 54.5% in DM group) and the restriction of social contact (47.16% in HD group and 47.6% in DM group), followed by limited access to a primary care physician and specialists (47.16% in HD group and 37.6% in DM group), and no family celebrations and trips.

### 3.4. Mental Disorders and the GAD-7 Scale

The results are given in Table 4 and include the statistics about mental disorders, the results of the GAD-7 scale, and the results of the GAD-7 scale with an additional question regarding anxiety related to COVID-19.

Depressive disorders were reported by 1 patient in the HD group and 17 subjects in the DM group. Similarly, other mental disorders were significantly more frequently reported in the DM group. Sleep problems were found in approximately 30% of patients. No differences were observed between the groups. Similarly, the percentage of patients administered hypnotic drugs was not different (approximately 20% in both groups). Also, approximately 20% of patients in both groups declared that the pandemic had negatively affected the quality of their sleep.

The mean score of the GAD-7 scale in the HD group did not differ according to gender [5 (2;10) in women and 4 (1;12) in men]. In the group of DM patients, a significant difference was observed between men and women, with women scoring higher compared to men [6 (2;10) vs. 2 (1;7)]. Similarly, when the score of the GAD-7 scale with an additional question was analyzed, significantly higher scores were obtained in women with DM and no gender difference was found in the HD group. In the DM group, the percentage of patients with a GAD-7 score > 5 was 39.43%, that with >10 was 16.57% and that above 15 was 4.57%; this is compared to 41.52%, 20.09% and 9.38%, respectively, in HD patients. No differences were found between the groups (Table 5).

The correlation analysis between the scores of the GAD-7 and GAD-7 with an additional question based on different parameters is given in Table 6 and Figure 1.

## 4. Discussion

Any form of confinement or isolation negatively affects the functioning of the population. Isolation means change, separation from relatives and a departure from daily routine. These factors are associated with increasing fear and anxiety, and can cause mental disorders. The new pandemic daily routine also brought changes to the functioning of the healthcare system. The patients who were mostly affected by these changes were those with chronic diseases.

Our study compared the functioning of two different groups of patients with chronic diseases during the pandemic. The first group included HD patients who, despite the pandemic, had to regularly visit the dialysis center, stay with other patients during transport and stay at the dialysis center. Due to the multimorbidity and immune disorders associated with CKD, this group was at an increased risk of SARS-CoV-2 infection.

Another group included DM patients who had been ill for many years, usually did not have direct contact with medical facilities and relied only on telephone consultations during the pandemic. Glycemic control deteriorated in most of these patients and lockdowns promoted weight gain. Many studies showed that poor glycemic control and obesity associated with type 2 DM were the most significant risk factors for severe COVID-19 infection [9,10,11,12].

The strength of this study is the patient group size and the fact that the questionnaires were distributed at the same time, which allowed us to conduct the study at a time when patients were exposed to the effects of the pandemic and quarantine in general to a similar level.

In the DM group, more patients were professionally active and a higher percentage of this group was adversely financially affected during the pandemic. The lack of impact the pandemic had on the financial situation of HD patients could be explained by the fact that they mostly did not work or were on disability living allowance.

The low percentage of patients eligible for kidney transplantation (12%) may be due to the limited access to diagnostic tests and other medical specialists experienced during this period, which was also confirmed in our study (nearly 50% of HD patients reported limited access to other specialists during the pandemic). According to the report on dialysis in Poland, the decrease in the number of patients eligible for kidney transplantation was associated with the pandemic, the increasing age of dialysis patients and multimorbidity, which adversely affected the reporting of new patients for transplantation [13].

In DM patients, there were no differences in BMI before the pandemic outbreak and after two years of the pandemic (approximately 31 kg/m^2^). DM patients were aware (>60%) that their glycemic control was significantly poorer during the pandemic (increase in HbA1c from 7.5% to 8.05%). The results of our study are in line with other recently published papers [14].

Although patients in both groups were similarly affected by COVID-19, DM patients required antibiotic treatment and oxygen therapy significantly more often. A more severe clinical course of infection may have been related to the significantly lower proportion of vaccinated DM patients in our study population.

The greatest difficulties for both groups associated with the pandemic were mask wearing and limited social contact, which significantly translated into mental disorders. HD patients did not report depressive disorders or other mental conditions. As in the case of DM patients, HD subjects declared sleep problems and used hypnotic drugs (>30%). Additionally, the pandemic worsened the quality of their sleep.

The GAD-7 questionnaire has become a widely used screening tool for assessing the prevalence of GAD in primary healthcare since 2006. Its usefulness, sensitivity and specificity are often emphasized. Of note, it allows consistent results to be obtained regardless of whether the patient completed the questionnaire himself/herself or did so with the assistance of the interviewer. Nevertheless, low rates of depressive disorders and other mental conditions in the HD group were observed. This may be because these patients were surrounded by the employees of the dialysis center and other patients, with whom they met several times a week. As a result, they may have been afraid of stigmatization and avoided additional questions or comments regarding the mental aspects of COVID-19, therefore not revealing anxiety symptoms. The different circumstances of the patients who completed the questionnaire were a main limitation of our study. In the future, more comfortable conditions for completing questionnaires should be provided, as data collection should not be performed during hemodialysis.

The GAD-7 scale itself has certain limitations as it is related to only one anxiety disorder, and other disorders, such as social phobia and post-traumatic stress disorder, are not included. The authors of the GAD-7 scale emphasize that it only detects a probable diagnosis that needs further confirmation. One of the limitations of this scale is the occurrence of differences in the interpretation of results between different ethnic groups.

In turn, the strength of our study is the comparison of two large populations of chronically ill patients with different diseases, excluding oncological conditions. To date, few data have been published on mental health and anxiety management during the pandemic with regard to these two groups.

We decided to choose these two groups of patients because both of them have chronic incurable illnesses that are associated with a significant risk of developing complications; hence, frequent contact with healthcare professionals is necessary.

Additionally, the populations of patients we selected had the worst prognosis in terms of the pandemic and were characterized by increased mortality, not due to COVID-19 infections, but due to cardiovascular complications. In these patient groups, the level of anxiety was expected to be amongst the highest.

When we analyzed the results of the GAD-7 scale, we found that in HD patients, the mean score did not differ according to gender and that its score did not even indicate the presence of mild anxiety. The percentage of patients who scored > 10 points was 20% and > 15 points was 9%.

Direct and frequent contact with healthcare professionals and the support that is given to patients during such visits may be the reason for the reduction in fear and anxiety in this group. In DM patients, a significant difference was found between men and women. On average, women scored 5 points higher, which indicated the presence of mild anxiety.

The psychological vulnerability of women has been described in several studies and it seems that the adverse impact of the pandemic and quarantine can be particularly seen in the female population [15,16]. The percentage of patients with a GAD-7 score > 10 was 16.57% in the DM group, which may indicate the presence of generalized anxiety. A score > 15 points, indicating severe anxiety disorder, was obtained by 4.57% of patients.

Next to the female gender in the DM group, the occurrence of sleep disorders and the use of hypnotic drugs were the factors that most significantly contributed to higher anxiety in both groups. In the HD group, difficulties related to transportation for dialysis, missed dialysis, or the postponement of the procedure significantly increased anxiety. A higher number of comorbidities in HD patients was also associated with significantly higher anxiety. The prevalence of anxiety disorders is estimated at 7.3% worldwide. Our results indicated that, despite the ongoing pandemic and many associated restrictions, HD patients presented with lower scores than those in the general population. DM patients presented with mild anxiety disorders, which was particularly visible in females.

A practical aspect of our study is related to the introduction of the regular assessment of anxiety disorders in the population of chronically ill patients, with special attention paid to the subpopulation of women since they are more susceptible to anxiety disorders.

## 5. Conclusions

Our study suggests that the psychological aspect of the treatment of somatic disorders must also be considered. In both groups, chronically ill patients reported anxiety disorders with similar frequency. In the DM group, more severe anxiety disorders were found in women. Mental disorders were significantly more prevalent in DM patients. It seems that HD patients coped better with the psychological aspects of pandemic-related stress and limitations.

Further studies should be conducted on the long-term impact of the COVID-19 pandemic on mental health in chronic patients and the dynamics of the changes in the post-pandemic period.

## 6. Take Home Message

The assessment of the mental health of patients is critical when somatic diseases are treated. Changes in medical care during the COVID-19 pandemic may have potentially increased the level of anxiety disorders. General practitioners and other medical professionals should be aware of an increased risk of anxiety disorders in such groups of patients, and should also actively search for disorders using rapid and easy screening tools, such as the GAD-7 scale. Medical and psychological support should be provided, which is particularly important during pandemics such as SARS-CoV-2. Instead of online medical visits, personal contact with healthcare personnel seems to have a positive effect on the well-being of chronically ill patients.

## Figures and Tables

**Figure 1 ijerph-20-06956-f001:**
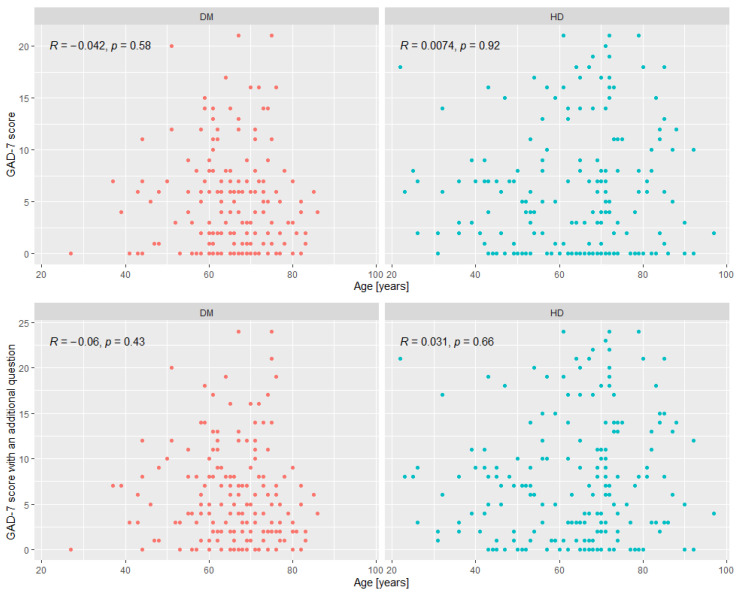
The *p* values for the Spearman’s rank correlation between the GAD-7 score and the GAD-7 score with an additional question and with age.

**Table 1 ijerph-20-06956-t001:** Sociodemographic data of the study participants.

	HD Patients (*n* = 244)	DM Patients (*n* = 175)	*p*— Test Chi^2^
Age (years)	63.7 ± 15.4	65.4 ± 10.3	*p* = 0.600
Sex (men, %)	125 (55.8)	70 (40)	*p* = 0.004
Marital status (married, %)	128 (57.1)	121 (69)	*p* = 0.001 * [prop test]
Housing status (living with family, %)	175 (78)	139 (79)	*p* = 0.826
Professionally active (yes, %)	28 (12.5)	48 (27)	*p* = 0.005
Financial deterioration during the pandemic (yes, %)	182 (81)	150 (85)	*p* = 0.034
Life changes due to the pandemic (yes, %)	153 (68.3)	118 (67)	*p* = 0.686

** proporsional test.*

**Table 2 ijerph-20-06956-t002:** Clinical data of the study participants.

	HD Patients (*n* = 244)	DM Patients (*n* = 175)	*p*
Duration of dialysis (months):	45 (19.2–84)		
Dialysis before the pandemic: (yes, %)	158 (70)		
Eligibility for transplant: (yes, %)	27 (12)		
Pandemic-related difficulties in the HD group (yes, %): - Missed dialysis - Transport delays/Postponement of surgery	15 (6.6) 26 (11.6)		
Duration of diabetes (years ± SD):		12.9 ± 8.31	
Treatment of diabetes (%):			
Oral		67 (38.3)	
Insulin		25 (14.3)	
Oral + insulin		83 (47.4)	
Comorbidities (%):			
Hypertension		142 (82)	
Hypercholesterolemia	192 (85.6)	105 (60)	*p* = 0.343
Ischemic artery disease	81 (36.2)	53 (29)	*p* < 0.0001
Chronic renal disease	156 (69.6)	22 (13)	*p* = 0.037
Number of comorbidities (%):			*p* = 0.038
0	29 (12.9)	15 (9)	
1	77 (34.3)	41 (27)	
2	74 (33)	72 (41)	
3	38 (16.9)	33 (19)	
≥4	6 (2.7)	7 (4)	
BMI: - Before the pandemic (kg/m^2^) - After two years (kg/m^2^)	27.7 (24.2–31.2) 26.8 (23.5–30.7) *p* < 0.0001	31.2 (27.7–36.1) 31.1 (27.7–36.2) *p* = 0.691	
HbA1c: - During the pandemic - At the time of the study		7.5 (6.4–9.2) 8.05 (6.6–10.1)	*p* < 0.0001
Glycemic control was poorer in the pandemic, according to the patient: (yes; %)		105 (63.6)	
Preferred visit type: (outpatient visit; %)	139 (62)	113 (65)	*p* = 0.986
Problems with obtaining prescriptions during the pandemic: (yes; %)	23 (10.2)	6 (3.0)	*p* = 0.001 or 3.138 (1.21–9.65) 95% CI
Difficulty contacting other specialists during the pandemic: (yes; %)	104 (46.4)	60 (34.2)	*p* = 0.014

**Table 3 ijerph-20-06956-t003:** COVID-19 status.

	HD Patients (*n* = 244)	DM Patients (*n* = 175)	*p* Test Chi^2^
Past COVID-19 infection: (yes; %)	160 (75.4)	125 (71.4)	*p* = 0.990
Severe course of COVID-19: (yes; %)	55 (24.5)	31 (18)	*p* = 0.544
Medications during the COVID-19 infection: (yes; %)			
- Oxygen therapy	28 (12.5)	44 (25.1)	*p* < 0.0001
- Antibiotics	19 (8.4)	29 (12.5)	*p* = 0.013
- Steroids	40 (17.8)	42 (24)	*p* = 0.407
Vaccination: (yes; %)	197 (87.9)	133 (77)	*p* = 0.013
Death of a relative due to COVID-19: (yes; %)	26 (11.6)	21 (12)	*p* = 0.990
Pandemic-related problems (%):			
- Restriction of contact	114 (47.16)	82 (47.6)	p = 0.990
- Mask-wearing	122 (50.9)	94 (54.5)	*p* = 0.990
- Closure of shops/restaurants	45 (18.8)	26 (15.2)	*p* = 0.721
- No travelling	82 (33.9)	49 (28.6)	*p* = 0.571
- No celebrations	73 (30.1)	42 (24.5)	*p* = 0.523
- Limited access to a primary care physician	105 (43.3)	80 (46.5)	*p* = 0.810
- Limited access to a medical specialist	114 (47.16)	64 (37.6)	*p* = 0.282

**Table 4 ijerph-20-06956-t004:** Mental disorders and the GAD-7 scale.

	HD (*n* = 244)	DM (*n* = 175)	*p* Test Chi^2^
Depressive disorders in the past (*n*, %)	1 (0.4)	17 (10)	*p* ≤ 0.0001
Other mental disorders (*n*, %)	1 (0.4)	11 (6)	*p* ≤ 0.0001
Sleep disorders	78 (34.8)	56 (32)	*p* = 0.990
Use of hypnotic drugs	45 (20)	38 (21.7)	*p* = 0.407
The pandemic had negatively affected their quality of sleep	38 (17)	35 (20)	*p* = 0.238
The GAD-7 scale:	4 (1–7)	5 (1–8)	
Women	3 (1–10)	1 (0–6)	
Men	*p* = 0.907	*p* = 0.001	
The GAD-7 scale with an additional question:	5 (1–10)	6 (1–10)	
Women	4 (1–12)	2 (1–7)	
Men	*p* = 0.638	*p* = 0.004	

**Table 5 ijerph-20-06956-t005:** The results of GAD-7.

	DM	HD	*p*
GAD-7 > 5 (%)	69 (39.43)	93 (41.52)	0.749
GAD-7 > 10 (%)	29 (16.57)	45 (20.09)	0.443
GAD-7 > 15 (%)	8 (4.57)	21 (9.38)	0.101

**Table 6 ijerph-20-06956-t006:** The correlation between the GAD-7 score and the GAD-7 score with an additional question, and the different sociodemographic data.

	GAD-7 Score		GAD-7 Score with an Additional Question on Anxiety Relating to COVID-19 Infection	
	HD	DM	HD	DM
Sex	*p* = 0.972	*p* = 0.001	*p* = 0.638	*p* = 0.004
Housing status (living on their own)	*p* = 0.344	*p* = 0.973	*p* = 0.308	*p* = 0.774
Marital status	*p* = 0.8	*p* = 0.521	*p* = 0.585	*p* = 0.240
History of COVID-19 infection	Median: 3 (0–8) 4.5 (0–12.2) *p* = 0.153 **	Median: 3 (0–8) 3 (0–6) *p* = 0.399 **	Median: 4 (1–10) 7 (1.75–14.2) *p* = 0.199 **	Median: 5 (2–10) 3.5 (1–7.75) *p* = 0.176 **
Use of hypnotic drugs	Median: 2 (0–7) 4 (0–9) *p* < 0.001 **		Median: 8 (3–8) 9 (6–18) *p* < 0.001 **	
Sleep disorders vs. anxiety level	Median: 2 (0–7) 3 (0–8) *p* < 0.001 **		Median: 7 (2–14) 9 (3–17) *p* < 0.001 **	
Missed dialysis	*p* = 0.001 ** 3 (0–8) 10 (7–15.5)		*p* = 0.002 ** 4 (1–10) 12 (8.5–16.50)	
Duration of diabetes		*p* = 0.115 * r = 0.119	*p* = 0.101 * r = 0.124	
Number of comorbidities	*p* < 0.001 * r = 0.035	*p* = 0.369 * r = 0.068	*p* < 0.001 * r = 0.021	*p* = 0.560 * r = 0.056

Abbreviations: HD, hemodialysis; DM, diabetes. *: Spearman’s correlation, **: Mann–Whitney test

## Data Availability

The data presented in this study are available on request from the corresponding author.

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
