# Peer review of "The Impact of the COVID-19 Pandemic on the Assessment of Healthcare and Anxiety Disorders in Patients with Chronic Diseases"

_ijerph, 2023, doi:10.3390/ijerph20206956_

Round 1

Reviewer 1 Report

There is not a clear operational definition of sdoh in the background/method or why this data was collected/though the rational for collecting can be inferred

Usual SDOH include access to food, transporation, as well as housing and economics - and I do not see this area discussed- thus the need for operationalizing SDOH for your study

In table 2 - it is a self reported improvement reported by individuals in A1C prior to and post covid? - Is all of the data self reported/ if so a sentence to clarify this would be helpful 

in the background/introductin line 61-62 provides information about those at risk of becoming diabetic and then not having a means to be diagnosed. . is this relevant to this manuscript and your study? 

I would infer from the beginning that HD patients had more social contact than DM patients- and not sure of the rational of why these two groups were selected when the daily approach to care is so very different - an explanation of the rational for selecting these two groups with know disparate approaches to care and social contacts would be helpful too 

Author Response

We would like to thank the Reviewers for all their invaluable comments and suggestions. All of them have helped us to clarify the manuscript and make significant changes that we hope will improve its form and content.

In our study, we evaluated SDOH - these aspects were covered in the first part of the questionnaire, in which we assessed socio-economic conditions, household conditions and the impact of the pandemic on the financial situation. The results are given in Table 1.

The results of HbA1c from the time before the pandemic were obtained from patient self-reporting, and after COVID-19 from medical records (patients hospitalized in the Department of Diabetology).

The introduction section described the risks associated with the pandemic, including delay in the diagnosis of the disease. In our study, we had no data on the delay in the diagnosis of diabetes mellitus.

Dialysis patients constitute a group that had more social contact in the pandemic, which was related to their treatment modality. We decided to choose these two groups of patients because both of them have chronic illnesses that are incurable, with a significant risk of developing complications, and hence frequent contact with healthcare professionals is necessary.

Additionally, the populations of patients we selected had the worst prognosis in terms of the pandemic and they were characterized by increased mortality not due to COVID-19 infections, but due to cardiovascular complications.  In these patient groups, the expected level of anxiety could be the highest.

We wanted to assess whether differences due to the treatment of these groups of patients could translate into the presence of anxiety disorders and other aspects of functioning during the COVID-19 pandemic.

Reviewer 2 Report

This is a cross-sectional study with the aim of investigating and comparing the impact of the COVID-19 pandemic on the functioning and mental health of two groups of patients, diabetics and patients on chronic dialysis. Despite the stated goal, the title of the study ("The impact of the COVID-19 pandemic on the quality of medical care in patients with chronic diseases") has absolutely nothing to do with this goal, nor with the written work! Furthermore, only the GAD-7 questionnaire was used in the research, which serves to assess exclusively anxiety disorders, so accordingly, this study does not investigate functioning or mental health as a whole, but only the impact of the COVID-19 pandemic on patient anxiety. I do not see any element in this study that investigates the quality of medical care that the title of the study indicates.

I disagree with the authors stating that this research does not require permission from the ethics committee. It is a questionnaire filled out by seriously ill people (patients on chronic dialysis and hospitalized diabetics) who have a risk of impaired ability to judge and understand the questions in the questionnaire, while they relate to the degree of anxiety. Mental health is always a sensitive topic and any similar research requires the approval of the ethics committee of both institutions where this research was conducted.

The study population and clinical data section belong to results, not methods. Some of these results are repeated in the text and tables. In Table 6, it is necessary to round the results to a reasonable number of decimal places (preferably three), because this type of data display has no special scientific value.

In conclusion, I don't see how the authors connected COVID-19 and the level of anxiety in the mentioned patients. This study has significant flaws in the design, implementation, and presentation of the results and conclusions derived from them. Unfortunately, this manuscript cannot be accepted for publication in this journal. If the authors want to investigate what they stated as the goal of the research, it is necessary to obtain permission from the ethics committee and completely change the design of the study, whereby it is necessary to include several different psychological questionnaires.

Author Response

Dear Sir/Madam,

We would like to thank the Reviewers for all their invaluable comments and suggestions. All of them have helped us to clarify the manuscript and make significant changes that we hope will improve its form and content.

In our study, we assessed various aspects of the functioning of chronically ill patients. I agree with the Reviewer that we used only one questionnaire assessing anxiety, which we developed and expanded with an additional question on anxiety related to COVID-19. Despite the fact that we did not use other ready-made validated scales, we conducted a detailed questionnaire among patients that assessed their socio-economic situation in the pandemic, housing conditions, morbidity, hospitalization and pandemic-related limitations. The questionnaire also included the questions about the co-occurrence of depressive and sleep disorders as well as the need for the use of hypnotic drugs.  

The study was questionnaire-based and hence the approval of the Bioethics Committee of the Medical University of Silesia was not required.  The proper statement from the Bioethics Committee has already been sent to the Publisher.

After the opinion from Reviewer 3, we highlighted the inclusion and exclusion criteria (inability to complete the questionnaire independently due to intellectual disorders).

According to the suggestion of the Reviewer, the study population and the clinical data section were moved to the Results section. Table 6 includes the results that were rounded.

We hope that after the correction and clarification of the information or changing the title, the paper will be suitable for publication. Would it better to change the tittle for for instance “The impact of the COVID-19 pandemic on assessment of healthcare and anxiety disorders in patients with chronic diseases”?

Reviewer 3 Report

1.       In this study, the inclusion and exclusion criteria are not clear.

2.       Line 180: the GAD-7 scale and the GAD-7 scale are repeated twice.

3.       In the result section, it is better to use the subgroups.

4.       The result section is hard to read for readers.

5.       It could be better to indicate the correlation with figures, not tables.

6.       The author has to explain the practical application of this study in the discussion section.

7.       Describe the limitation and straightness and suggestions for future study.

8.       What are the home messages?

Author Response

Dear Sir/Madam,

We would like to thank the Reviewers for all their invaluable comments and suggestions. All of them have helped us to clarify the manuscript and make significant changes that we hope will improve its form and content.

Please find the responses that were included in the manuscript:

  1. Line 180: It was not repeated twice because we analyzed the “GAD-7” and the “GAD-7 with an additional question".
  2. I used the subgroups in the Results section.
  3. I tried to simplify this section.
  4. I indicated the correlation with figures.

5,6,7 The information was added to the main text.

Round 2

Reviewer 2 Report

Dear researches, 

some parts of this manuscript were, indeed, improved. But, I still have many concerns about the study. 

Results: How many patients with type 2 diabetes were in the hemodialysis group? T2DM is one of the leading causes od chronic kidney disease. Also, on which patients who have T2DM do you reffer, because you can have multiple confounders in analysis of heterogenous group of patients like this. For example, insulin-dependent T2DM patients and those on peroral therapy (especially with only metformin in therapy) have completely different fears, perception of the diseases (and therefore anxiety levels) and that should be taken into account. As I mentioned earlier, this study design cannot answer the aim clearly.

This study used GAD-7 questionnaire. I strongly disagree that the ethical approval for this study is not required. You can use this questionnaire freely, without the permission to reproduce, display or distrubute the questionnaire but if you're using it in a research on human subjects the permission is needed. Not only because of this questionnaire but also because of the study design.  

Results:  

Line 211 - 213. These results are not means!

Table 1 is a mix of many various variables shown as means, standard deviations, medians (interquartile ranges?? - it is not clear because they are usualy shown with "-", not ";"). Also, please fix the number of decimal places (it should be equal everywhere).

Table 6 - p value cannot be 0. It is a probability! These p-values have a little scientific value when shown without actual correlation ranks. "Use of hypnotic drugs", "Sleep disorders vs. anxiety level", "Missed dyalisis" and "Duration of diabetes" are shown differently from the other variables without any explanation which makes it difficult to interpretate.

As strenght of this study, authors mentioned that they excluded oncological conditions, but this is not stated in exclusion criteria earlier in methods section. 

In conclusion, this study still has a questionable design, some possible systemic errors and it's scientific contribution is extremely low. Maybe the authors can increase it by adding a control group to this study. Unfortunately, I still think that this manuscript is not suitable for publication in this Jorunal.

Author Response

We would like to thank the Reviewer for all the invaluable comments and suggestions. All of them have helped us to clarify the manuscript and make significant changes that we hope will improve its form and content.

Please find the responses that were included in the manuscript:

- How many patients with type 2 diabetes were in the hemodialysis group?

There were 73 patients (32%) with T2DM in the hemodialysis group. We analyzed them together with all HD patients and evaluated the impact of the COVID-19 pandemic on the functioning and their mental health.

-

Also, on which patients who have T2DM do you 1tren, because you can have multiple confounders in analysis of heterogenous group of patients like this. For example, insulin-dependent T2DM patients and those on peroral therapy (especially with only metformin in therapy) have completely different fears, perception of the diseases (and therefore anxiety levels) and that should be taken into account. As I mentioned earlier,

this study design cannot answer the aim clearly.

In the group of DM patients, we excluded patients with newly diagnosed DM. We analyzed only patients who required intensification of DM treatment (for this reason they were hospitalized in the Department of Diabetology) and most of them were treated with insulin or

insulin with oral medications (61.7% in total).

The rest of the group used at least two oral

medications but required intensification of treatment.

- This study used GAD-7 questionnaire. I strongly disagree that the ethical approval for this study is not required. You can use this questionnaire freely, without the permission to reproduce, display or 1trength1e the questionnaire but if you’re using it in a research on human subjects the permission is needed. Not only because of this questionnaire but also because of the study design.

The study was questionnaire-based and the Bioethics Committee of the Medical University of Silesia was informed about the study. The proper statement from the Bioethics Committee has already been sent to the Publisher.

- Line 211 – 213. These results are not means!

The results were corrected as below:

Both groups had similar opinions about the pandemic-related inconvenience, the most significant limitations being mask-wearing (50.9% in HD group and 54.5% in DM group) and restriction of social contact (47.16% in HD group and 47.6% in DM group), followed by limited access to a primary care physician and specialists (47.16% in HD group and 37.6% in DM group), no family celebrations and trips.

- Table 1 is a mix of many various variables shown as means, standard deviations, medians (interquartile ranges?? – it is not clear because they are 1trengt shown with “-

“, not “;”). Also, please fix the number of decimal places (it should be equal everywhere).

Table 1 is a mix of many various variables because different tests were used.

All interquartile ranges were corrected according to your suggestion. The number of decimal places was also corrected.

- Table 6 – p value cannot be 0. It is a probability! These p-values have a little scientific value when shown without actual correlation ranks. “Use of hypnotic drugs”, “Sleep disorders vs. anxiety level”, “Missed dyalisis” and “Duration of diabetes” are shown differently from the other variables without any explanation which makes it difficult to interpretate.

p value was corrected and the missing data were completed. The description of table 6 was verified.

- As strength of this study, authors mentioned that they excluded oncological conditions, but this is not stated in exclusion criteria earlier in methods section.

We completed the exclusion criteria.

- In conclusion, this study still has a questionable design, some possible systemic errors and it's scientific contribution is extremely low. Maybe the authors can increase it by adding a control group to this study. Unfortunately, I still think that this manuscript is not suitable for publication in Journal.

We plan to conduct further research on the long-term impact of the pandemic on the functioning of patients with chronic diseases.

Our study was a real-life study and the control group is not required because we assessed the behavior of the specific population groups.

We truly hope that the changes introduced in accordance with the Reviewers’ suggestions make our manuscript suitable for publication.

Thank you for your time and consideration.

Reviewer 3 Report

It is ok.

Author Response

I enclose the last version of my manuscript.
